# Modulation of the Circadian Rhythm and Oxidative Stress as Molecular Targets to Improve Vascular Dementia: A Pharmacological Perspective

**DOI:** 10.3390/ijms25084401

**Published:** 2024-04-16

**Authors:** Walter Ángel Trujillo-Rangel, Sofía Acuña-Vaca, Danna Jocelyn Padilla-Ponce, Florencia Guillermina García-Mercado, Blanca Miriam Torres-Mendoza, Fermín P. Pacheco-Moises, Martha Escoto-Delgadillo, Leonel García-Benavides, Daniela L. C. Delgado-Lara

**Affiliations:** 1Departamento de Ciencias Biomédicas, Centro Universitario de Tonalá, Universidad de Guadalajara, Av. Nuevo Periférico No. 555, Ejido San José Tateposco, Tonalá 45425, Jalisco, Mexico; walter.trujillo@edu.uag.mx (W.Á.T.-R.); danna.padilla6381@alumnos.udg.mx (D.J.P.-P.); florencia.garcia5065@alumnos.udg.mx (F.G.G.-M.); drleonelgb@hotmail.com (L.G.-B.); 2Departamento de Formación Universitaria Ciencias de la Salud, Universidad Autónoma de Guadalajara, Av. Patria 1201, Lomas del Valle, Zapopan 45129, Jalisco, Mexico; sofia.acuna@edu.uag.mx; 3División de Neurociencias, Centro de Investigación Biomédica de Occidente, Instituto Mexicano del Seguro Social, Sierra Mojada 800, Colonia Independencia, Guadalajara 44340, Jalisco, Mexico; bltorres8@gmail.com (B.M.T.-M.); martha.escotod@gmail.com (M.E.-D.); 4Departamento de Disciplinas Filosófico, Metodológicas e Instrumentales, Centro Universitario de Ciencias de la Salud, Universidad de Guadalajara, Colonia Independencia, Guadalajara 44340, Jalisco, Mexico; 5Departamento de Química, Centro Universitario de Ciencias Exactas e Ingenierías, Universidad de Guadalajara, Blvd. Marcelino García Barragán No. 1421, Guadalajara 44430, Jalisco, Mexico; ferminpacheco@hotmail.com; 6Centro Universitario de Ciencias Biológicas y Agropecuarias, Universidad de Guadalajara, Camino Ramón Padilla Sánchez No. 2100, Zapopan 45200, Jalisco, Mexico

**Keywords:** vascular dementia, oxidative stress, circadian rhythm, pharmacology

## Abstract

The circadian rhythms generated by the master biological clock located in the brain’s hypothalamus influence central physiological processes. At the molecular level, a core set of clock genes interact to form transcription–translation feedback loops that provide the molecular basis of the circadian rhythm. In animal models of disease, a desynchronization of clock genes in peripheral tissues with the central master clock has been detected. Interestingly, patients with vascular dementia have sleep disorders and irregular sleep patterns. These alterations in circadian rhythms impact hormonal levels, cardiovascular health (including blood pressure regulation and blood vessel function), and the pattern of expression and activity of antioxidant enzymes. Additionally, oxidative stress in vascular dementia can arise from ischemia-reperfusion injury, amyloid-beta production, the abnormal phosphorylation of tau protein, and alterations in neurotransmitters, among others. Several signaling pathways are involved in the pathogenesis of vascular dementia. While the precise mechanisms linking circadian rhythms and vascular dementia are still being studied, there is evidence to suggest that maintaining healthy sleep patterns and supporting proper circadian rhythm function may be important for reducing the risk of vascular dementia. Here, we reviewed the main mechanisms of action of molecular targets related to the circadian cycle and oxidative stress in vascular dementia.

## 1. Introduction

Vascular dementia is considered the second most common subtype of dementia after Alzheimer’s disease (AD). It manifests in around 15 to 20% of dementia cases in North America and Europe, while it accounts for 30% of cases in Asia [1]. The prevalence of vascular dementia is higher in men than in women under 80 years old; yet, as age increases, this relationship reverses [2]. The main clinical sign of vascular dementia is executive dysfunction (slowed information processing, impairment in the ability to shift from one task to another, and deficits in the ability to maintain and manipulate information), followed by memory and language deficits [3]. The origin of vascular dementia stems from a variety of factors leading to damage to cerebral blood vessels, consequently disrupting the flow of blood and oxygen and resulting in hypoperfusion. Additionally, endothelial damage may precipitate neurovascular dysfunction, heightened permeability of the blood–brain barrier, and microvascular thrombosis [1]. 

The risk of developing vascular dementia increases after a stroke, but this depends on the size and number of strokes a person has had and the regions of the brain affected [4,5]. Similarly, vascular diseases such as atherosclerosis or high blood pressure contribute to the development of vascular dementia. Variations in blood pressure and heart rate are well-known circadian rhythms of physiology. An abnormal circadian rhythm of blood pressure, in addition to being a predictor of cardiovascular and cerebrovascular diseases, may lead to an increased risk of cognitive dysfunction [4].

An abnormal circadian rhythm can alter markers of oxidative stress and vascular endothelial function, which, in turn, will increase the risk of adverse cardiovascular events in susceptible individuals [6]. The discovery of the relationship between the circadian rhythm and vascular dementia opens a new avenue for exploring molecular targets and, accordingly, the development of pharmacological interventions to potentially delay the progression of the disease. Therefore, the objective of this literature review is to understand how the circadian rhythm and oxidative stress are related to vascular dementia, as well as to identify circadian cycle markers associated with oxidative stress to enhance the management of vascular dementia.

## 2. Circadian Rhythm and Vascular Dementia

Circadian rhythms are physiological and behavioral cycles with a periodicity of 24 h, generated by an internal biological clock, the suprachiasmatic nucleus (SCN), located in the anterior hypothalamus [7]. The SCN receives information from external zeitgebers and uses that information to synchronize physiology and behavior with the 24 h rotation of the earth. Light is the most crucial environmental zeitgeber. Other external signals, such as physical or motor activity and meal timing, also facilitate the synchronization of circadian rhythms [7,8,9]. 

In general, when exposed to a light stimulus, light is absorbed by the ganglion cells of the retina that contain melanopsin (photoreceptor cells) [10]. The stimuli travel through glutamate signaling along the retinohypothalamic tract and finally to the SCN [11,12]. The SCN transmits its circadian signal to peripheral tissues through neural and hormonal mechanisms, synchronizing circadian oscillations throughout the body. Endocrine parameters, such as melatonin and cortisol levels, as well as central body temperature and blood pressure (BP), can be used as surrogate markers of the central clock in vivo, as the SCN generates their rhythmic production [7,13]. 

At the molecular level, on a regular basis, a basic set of clock genes, including *CLOCK* (*circadian locomotor output cycles kaput*), *BMAL1* (*brain and muscle ARNT-like protein 1*), the *period* gene (*PER1*, *PER2*, and *PER3*), and the *cryptochrome* gene (*CRY1* and *CRY2*), interact to form transcription–translation feedback loops that provide the molecular basis of the circadian rhythm [7,9]. A key pathway involves the proteins BMAL1 and CLOCK, which form a heterodimer to regulate the expression of clock genes, leading to the circadian response of the SCN. The CLOCK/BMAL1 complex activates the expression of the *PER* and *CRY* genes, whose products (PER and CRY) act as harmful elements in the pathway and inhibit the activity of BMAL1/CLOCK (and thus their expression). In a separate stabilization pathway, the *Rev-Erbα* and the *orphan retinoic receptor* genes adjust the oscillations generated by the main pathway. Unlike other PER proteins, PER2 can directly interact with Rev-Erbα to regulate the expression of BMAL1 [14,15,16] (Figure 1).

In patients with neurodegenerative diseases, such as Huntington’s disease, Parkinson’s disease, and Alzheimer’s disease, compared to healthy individuals, impaired circadian rhythms have been found. Generally, the average production of melatonin decreases or the amplitude of its secretion decreases; cortisol levels rise in the morning and/or their amplitude is altered; and body temperature shows a reduction in its amplitude or a phase delay in its rhythm, and blood pressure increases [13,17]. Regarding the expression patterns of clock genes, there are few studies measuring them in individuals with neurodegenerative diseases [18,19,20,21], but it has been seen that, with age, their expression in the cortex becomes dysregulated, either showing a phase shift or loss of rhythmicity. Sometimes, the arrhythmic genes of young adults become rhythmic in older adults. This type of complexity in the expression of clock genes occurs in patients with neurodegenerative diseases [17,22] (Table 1).

There is a lack of studies evaluating peripheral and molecular markers in patients with vascular dementia; however, in patients with cerebrovascular disease, it has been described that melatonin production decreases. As melatonin levels decrease, there is a higher probability of experiencing a heart attack and sleep disorders [28]. Regarding cortisol levels, it has been emphasized that excessive levels of cortisol are associated with an increased risk of cardiovascular events such as acute coronary syndrome, arrhythmias, sudden cardiac death, and heart attack [29]. With regards to body temperature, patients with multi-infarct dementia have shown a decrease in the amplitude of their rhythm compared to patients with Alzheimer’s-type dementia [30]. Patients with dementia have higher systolic blood pressure (SBP) than healthy individuals and an abnormal circadian rhythm. Specifically, patients with vascular dementia have higher SBP and diastolic blood pressure (DBP) than patients with AD and healthy individuals, both at night and during the day [31].

However, in various rat models with cardiovascular diseases, clock genes have been measured in peripheral tissues, revealing a lack of synchronization with the central clock, the SCN. This loss of synchronization may exacerbate the incidence of cardiovascular diseases such as hypertension, myocardial infarction, cardiac hypertrophy, atherosclerosis, and angiogenesis [32], as well as metabolic disorders like hepatic steatosis, obesity, metabolic syndrome, and diabetes mellitus [33].

It is important to note that Alzheimer’s-type dementia is characterized by the accumulation of beta-amyloid plaques and neurofibrillary tangles in the brain, leading to neuronal injury and death and resulting in memory loss and cognitive decline. Symptoms of Alzheimer’s include progressive memory loss, cognitive deficits affecting various domains, and noncognitive symptoms such as depression, psychotic symptoms, and behavioral changes. Personality changes and hallucinations may occur as the disease progresses, ultimately leading to severe impairment and loss of basic functions. While memory loss is a prominent feature in Alzheimer’s, vascular dementia symptoms can vary widely depending on the affected brain regions and severity of blood vessel damage, often involving executive function deficits and sudden cognitive changes after strokes. Additionally, Alzheimer’s typically progresses gradually, while vascular dementia progression may vary depending on the occurrence and severity of vascular events [34].

Several studies agree that the disruption of the circadian rhythm, whether in terms of peripheral or molecular markers, can contribute to the development and progression of dementia. Low levels of melatonin and high levels of cortisol in various neurodegenerative diseases have been associated with a higher incidence of dementia [29]. For example, in Alzheimer’s disease, the loss of daily rhythmic patterns that modify the expression of the *BMAL1* gene correlates with an increased segregation of the TAU protein, more significant cognitive impairment, and nocturnal awakenings [35]. Similarly, in a mouse model of Huntington’s disease, it was observed that, as the disease progresses, there is more significant variability in the cycles of body temperature, and a higher amplitude in its rhythm. In patients with Parkinson’s disease, it was noted that the dysregulation of clock genes, specifically the expression levels of the *BMAL1* gene, is associated with the severity of the disease and more motor problems [19,36]. The disruption of the circadian blood pressure cycle can also affect the cognitive function of patients with small vessel disease; healthy individuals experience a nighttime decrease of 10–20% in blood pressure, known as the dipping pattern. However, those with a reverse-dipping pattern (decrease in BP less than 0%) or a non-dipping pattern (decrease in BP of between 0 and 10%) have a higher risk of cognitive dysfunction [4].

Therefore, this disruption of the circadian cycle can influence various physiological processes, including sleep–wake cycles and cognitive performance, affecting the primary symptoms of each disease and overall quality of life. It has been reported that patients with sleep disorders have a higher risk of dementia compared to those without disorders. Specifically, insomnia is associated with Alzheimer’s-type dementia, and sleep respiratory disorders are associated with all types of dementia, including vascular dementia [16]. Risk factors that predispose to dementia include older age, female gender, family history, severe traumatic brain injury, and genetic mutations. However, there are also modifiable risk factors that could prevent or delay dementia by up to 40%, including depression, physical activity, obesity, tobacco use, diabetes mellitus, and untreated hypertension in midlife [37]. Recently, it has been proposed that sleep disorders could also be a modifiable risk factor for dementia, and that sleeping less than six hours at the age of 50 to 60 is associated with a higher risk of dementia [38,39,40,41].

## 3. Oxidative Stress and Vascular Dementia

Oxidative stress is a pathological process in which reactive oxygen species (ROS) and free radicals accumulate in the body, causing damage to cellular structures, including deoxyribonucleic acid (DNA), proteins, and lipids. The production of radical species is inevitable, and, indeed, some studies argue that it is essential, as they are generated in normal metabolism, originating from the electron transport chain in mitochondria by certain enzymes such as reduced nicotinamide adenine dinucleotide phosphate (NADPH) oxidase, and by phagocytic cells in the innate immune response [42].

Our understanding of the role of radical species in cellular biology has evolved considerably, revealing their crucial role as necessary modulators in a variety of intracellular signaling processes. These species, including free radicals such as singlet oxygen and hydroxyl radicals, are involved in the fine modulation of key signaling pathways such as the phosphorylation cascade and gene expression. For example, it has been demonstrated that free radicals participate in the activation of transcription factors and the regulation of the immune response. Additionally, these species are implicated in the modulation of apoptosis, the antioxidant response, and calcium homeostasis, among other fundamental cellular processes. As research in this area deepens, a landscape emerges where radical species appear as essential regulators in the cellular signaling network, whose dysfunction may contribute to various diseases and pathological conditions. However, the prolonged, uncontrolled production of radicals can lead to an environment of oxidative stress and subsequent damage to biomolecules [42,43].

To keep the balance between prooxidants and antioxidants, there is an extensive antioxidant system: endogenous enzymes such as superoxide dismutase (SOD), glutathione peroxidase (GPX), and catalase are present to eliminate superoxide, hydrogen peroxide, and singlet oxygen, respectively. These enzymes serve as antioxidants in a balanced diet. Additionally, there are repair and renewal mechanisms. Only when the production of ROS overwhelms these defenses can harmful effects occur in biomolecules. It is estimated that a single cell produces between 1 and 3 billion radical species each day, and it is clear from these figures that these antioxidant and repair mechanisms are essential for regulating cellular homeostasis [44].

One of the main risk factors for vascular dementia is atherosclerosis, which is intimately related to an increase in oxidative stress, leading to an increase in atherosclerotic plaque, and likely playing a significant role in the development of vascular dementia [45]. A critical factor that promotes the development of atherosclerotic plaques is the altered regulation of nitric oxide (NO). Nitric oxide is derived from the conversion of L-arginine to L-citrulline in the presence of cofactors, including tetrahydrobiopterin (BH4), through the action of nitric oxide synthase (NOS). There are three isoforms of NOS, namely nNOS (neuronal), iNOS (inducible), and eNOS (endothelial), each playing specific roles [45].

Elevated levels of oxidative stress can mediate the deregulation of NO production, as the bioavailability of NO is reduced due to the increased production of oxidants. Firstly, ROS can scavenge NO to form peroxynitrite, a highly potent radical species, thereby reducing NO circulation. It has been demonstrated that peroxynitrite formation occurs in the microvascular endothelial cells in rats, confirming that endothelial cells are a significant site susceptible to dysfunction. Secondly, the oxidation of BH4 leads to the uncoupling of endothelial nitric oxide synthase (eNOS) from NO production, and subsequently, superoxide is formed [45].

In a study conducted on the brain and cerebrospinal fluid (CSF) of individuals with AD, elderly individuals and younger controls were compared. The results indicated that both the AD group and the elderly group experienced an increase in ROS levels and signs of oxidation in proteins (carbonyl groups), lipids (malondialdehyde (MDA) and 4-hydroxynonenal), and DNA (8-hydroxydeoxyguanine (8-oxo-dG)). Additionally, a decrease in the activity of the enzyme glutamine synthetase was detected, resulting in a reduction in the ability to eliminate glutamate, and an increase in its toxic potential. Amyloid beta (Aβ) can induce oxidation through various pathways, affecting the p75 receptor and the receptor for advanced glycation end products (RAGE), generating a significant influx of calcium ions (Ca^2+^) and triggering caspase activation, leading to the process of cellular apoptosis [16].

In the context of vascular dementia, oxidative stress plays a crucial role in vascular dysfunction, inflammation, and neuronal damage. Several studies have emphasized that the regulation of oxidative stress is fundamental in the management of this neurodegenerative disease [13]. Previously, an increase in free radical production has been associated with cerebral ischemia, a risk factor for vascular dementia. Although studies measuring oxidative stress markers specifically in vascular dementia are limited, there are data supporting the role of oxidative stress in this form of dementia. Several studies have focused on investigating oxidative stress markers in the circulation [46].

Sinclair and colleagues demonstrated reduced levels of vitamin C in the plasma of individuals with vascular dementia compared to age-matched controls and, later, it was shown that α-tocopherol levels were also reduced in vascular dementia [47]. Other studies have also reported a reduction in a wide range of antioxidant micronutrients including α-tocopherol. Casado and colleagues demonstrated an increase in MDA in vascular dementia compared to healthy subjects of the same age. Additionally, oxidative damage to DNA, measured by 8-oxo-2′-deoxyguanosine and levels of 8-oxo guanine, was elevated in the CSF and urine of individuals with vascular dementia [48]. From this perspective, the data suggest that, at least systemically, the oxidative stress status is altered in patients with vascular dementia. 

Moreover, oxidative stress has been closely linked to the progression of vascular dementia. The process, characterized by an imbalance between the production of reactive oxygen species and the body’s antioxidant capacity, can trigger cellular damage and contribute to the cognitive degeneration associated with the disease. Studies have shown that elevated levels of oxidative stress markers, such as lipid and protein oxidation products, are correlated with an increased severity of vascular dementia symptoms and an acceleration in cognitive function loss [49]. This finding suggests that oxidative stress is not only involved in the etiology of vascular dementia but may also act as a key contributing factor to its progression [43]. Therefore, understanding and addressing oxidative stress could provide new therapeutic opportunities to delay the progression of vascular dementia and improve the quality of life for affected patients.

### Mitochondrial Dysfunction and Neuronal Cell Death in Vascular Dementia

Increased oxidative stress plays a pivotal role in the pathogenesis of vascular dementia, primarily affecting mitochondria and leading to neuronal cell death. The intricate interplay between oxidative stress, mitochondrial dysfunction, and neuroinflammation contributes significantly to the progression of this debilitating condition [50]. Mitochondria, often referred to as the powerhouse of cells, are particularly vulnerable to oxidative stress due to their high metabolic activity and abundant production of ROS as natural byproducts of respiration [51]. When the balance between ROS production and antioxidant defense mechanisms is disrupted, excessive ROS accumulation leads to the oxidative damage of mitochondrial DNA, proteins, and lipids. This damage impairs mitochondrial function, resulting in decreased adenosine triphosphate (ATP) production, disrupted calcium homeostasis, and compromised cellular bioenergetics [52]. 

In vascular dementia, chronic cerebral hypoperfusion and ischemic insults exacerbate oxidative stress, further compromising mitochondrial integrity and function. Impaired mitochondrial respiration and ATP synthesis not only compromise neuronal viability but also contribute to the activation of apoptotic pathways, ultimately leading to neuronal cell death [53]. Moreover, dysfunctional mitochondria release pro-apoptotic factors into the cytoplasm, amplifying neuronal damage and neurodegeneration [54].

Neuronal cell death triggered by oxidative stress and mitochondrial dysfunction initiates a cascade of inflammatory responses within the brain. Microglia, the resident immune cells of the central nervous system, become activated in response to neuronal injury, releasing pro-inflammatory cytokines and chemokines. This neuroinflammatory response exacerbates tissue damage, perpetuating a vicious cycle of oxidative stress, mitochondrial dysfunction, and neurodegeneration [54].

The underlying mechanisms linking oxidative stress, mitochondrial dysfunction, and neuroinflammation in vascular dementia are multifaceted. The activation of transcription factors such as NF-κB and nuclear factor 2 related to erythroid (Nrf2) mediates the expression of genes involved in inflammation and oxidative stress responses, respectively. Additionally, impaired mitochondrial quality control mechanisms, including mitophagy and mitochondrial dynamics, contribute to the accumulation of damaged mitochondria and exacerbate neuronal dysfunction [55].

In summary, increased oxidative stress in vascular dementia precipitates mitochondrial dysfunction, neuronal cell death, and neuroinflammation, collectively driving disease progression. Understanding the intricate interplay between these processes is crucial for the development of targeted therapeutic strategies aimed at mitigating oxidative damage and preserving neuronal function in individuals afflicted with vascular dementia.

## 4. Molecular Targets Related to the Circadian Rhythm and Oxidative Stress in Vascular Dementia

Oxidative stress is closely related to the circadian rhythm. The disruption of the circadian rhythm can lead to the overproduction of free radicals, resulting in oxidative damage to cellular components [56]. Moreover, there is a bidirectional relationship between circadian rhythms and oxidative stress, with disruptions to circadian rhythms affecting redox biology and the circadian pattern of expression and activity of antioxidant enzymes [57]. Antioxidant systems come into play to counteract ROS at low levels. Primarily, NADPH, produced through the pentose phosphate pathway, serves as a redox cofactor for antioxidant enzymes such as glutathione reductase (GR) and NADPH oxidase [29].

It has been mentioned that increased oxidative stress is critical in neurodegeneration due to the disruption of the circadian cycle, where the redox system impacts the functioning of the molecular clock while clock proteins monitor redox homeostasis [12]. The BMAL1:CLOCK dimer binding to DNA depends on its relationship with NADH and NADPH. The suppression of the pentose phosphate pathway results in the loss of NADPH, causing oxidative stress, and consequently activating Nrf2, increasing the binding of BMAL1:CLOCK to DNA [58]. *BMAL1* triggers the transcription of NADPH quinone dehydrogenase and aldehyde dehydrogenase 2 (QR2) in the brain [59]. 

The *PER* gene, another of the most important clock genes in the negative feedback loop of the circadian cycle, has been associated with antioxidant effects, particularly the *PER2* subtype, and with neuroprotective effects against oxidative damage. Furthermore, PER cleavage can accelerate neurodegeneration in dementias [12]. Some proteins, such as Aβ present in Alzheimer’s disease, alter the expression of *PER1* and *PER2* in the SCN, and the degradation of BMAL1, deregulating the circadian rhythm [60]. On the other hand, altered levels of *PER* gene expression have been correlated with cardiac dysfunction in patients with vascular dementia, and with their comorbidities such as atherosclerosis [12].

Furthermore, a group of transcription factors, forkhead-box-O (FOXO), are regulated by ROS and insulin through the Jun N-terminal kinase (JNK) and phosphoinositide 3-kinase (PI3K) pathways and stimulate *CLOCK* transcription. The FOXO3 subtype stimulates to sirtuin 1 (SIRT1) through the nutrient-sensing pathway of protein 53. Other proteins, such as NAMPT and gene 14 related to autophagy, are essential for lipid metabolism and autophagy. The mRNA of NAMPT, and the autophagy-associated gene 14 (ATG14), are crucial for lipid metabolism and autophagy. Additionally, both NAMPT and ATG14 mRNA show variations throughout the day according to the circadian biological clock, and their respective genes have promoter segments for FOXO proteins and the CLOCK: BMAL1 combination [61]. The functions of core clock genes are also controlled by post-translational modifications that follow a rhythmic pattern. As an example, the different variants of the genes belonging to the casein kinase 1 (CK1) family phosphorylate the PER protein in a time-of-day-dependent manner, thus ensuring the proper synchronization of the biological clock [12] Figure 2.

In addition to the pathways already mentioned, ubiquitination, phosphorylation, and acetylation modifications are essential in the regulation of the molecular clock. In mammals, two isoforms of CK1 (δ and ε) are involved in the modulation of circadian rhythms. The CK1 ε/δ phosphorylates and degrades the PER protein [62]. Another nuclear protein, βTrCP, also degrades PER2 by ubiquitination. The adaptation of activity in response to changes in temperature is a characteristic that has been maintained throughout evolution, and it seems to be related to changes that affect proteins in stages after their synthesis. It was suggested that CK1 isoforms are not influenced by thermal variations, which could make them responsible for the adaptability of the activity to temperature through their impact on the phosphorescent mechanism [61]. In the hippocampus of brains affected by Alzheimer’s disease, Ck1δ mRNA levels increase 24-fold. The production of the corresponding proteins follows a pattern like that of mRNA expression, and these proteins are found in the exact location of senile plaques [15].

Studies have shown that ROS production increases during certain phases of the circadian cycle, which may be influenced by metabolic activity and exposure to environmental factors. It has been observed that ROS generation in the brain peaks during waking hours and decreases during sleep, suggesting a possible interaction between the circadian cycle and oxidative stress in the context of brain function [63] (Figure 3). 

As previously mentioned, the peripheral and molecular markers of the circadian rhythm in neurodegenerative diseases are altered and can have significant consequences on cognitive function and the development of vascular dementia. For example, melatonin, in addition to regulating the sleep–wake cycle, has antioxidant properties and plays an essential role in protecting the brain against oxidative stress and neuroinflammation. Decreased melatonin levels in patients with cerebrovascular disease may exacerbate oxidative damage and contribute to the cognitive impairment associated with vascular dementia [19,43]. 

Similarly, the dysregulation of the hypothalamic–pituitary–adrenal (HPA) axis, which regulates cortisol release in response to stress, may also influence the pathogenesis of vascular dementia. It has been shown that elevated levels of cortisol, mainly when produced chronically due to circadian rhythm dysfunction, can damage the hippocampus and other brain areas involved in cognition and memory [29,58]. 

Autophagy pathways are also affected in dementia, as there is a decrease in the expression of Beclin 1, which initiates the autophagy process and is also linked to the sequestration and elimination of intracellular deposits [9]. Removing toxic compounds within cells may be crucial in preserving memory and cognitive ability. Autophagy, as a component of a process of planned cellular self-destruction, is connected to oxidative stress and the circadian clock [64]. Cell preservation relies on the stimulation of autophagy using proteins related to the circadian rhythm during stroke episodes. The inability of the circadian protein PER1 to function correctly can increase damage to brain tissue due to lack of oxygen [65]. 

Finally, endothelial dysfunction, a common feature of vascular dementia, is closely linked to circadian cycle regulation. The endothelium, lining the blood vessels, is crucial for maintaining vascular function and blood flow. Studies suggest that circadian cycle dysregulation can exacerbate endothelial dysfunction, implying that correcting circadian rhythm disorders might benefit vascular health and potentially prevent vascular dementia [66]. Circadian rhythms control the expression of genes involved in vascular homeostasis and endothelial function. Key genes include CCN3, α-adducin (G460W ADD1), β2-adrenoceptor (+46G/A ADRB2), endothelin-1 (Lys198Asn EDN1), G β3 protein subunit (G/A GNB3), cytochrome 3A5 (+6986G/A CYP3A5), neuropilin-1 (NRP1), and C-type natriuretic peptide (CNP). These genes are essential for maintaining vascular balance, regulating endothelial function, and preventing endothelial dysfunction [67,68]. In summary, correcting circadian cycle disorders may positively impact vascular health and help prevent vascular dementia by regulating endothelial function and maintaining vascular homeostasis.

## 5. Mechanism of Action: Molecular Targets Related to the Circadian Cycle and Oxidative Stress in Vascular Dementia

Given the observed correlation between circadian rhythm disruption and heightened susceptibility to vascular dementia, the significance of clock genes in modulating lipid metabolism and oxidative stress within the pathogenesis of this condition is underscored, as previously indicated [69]. The mechanisms of action of molecular targets in modulating the circadian cycle and oxidative stress play a crucial role in the search for practical pharmacological approaches to improve vascular dementia.

### 5.1. Pharmacological Interventions in the Circadian Cycle and Oxidative Stress

Several pharmacological compounds have been investigated in relation to modulating the circadian cycle and reducing oxidative stress in vascular dementia. These include agents that directly regulate the circadian clock, such as melatonin receptor agonists, as well as compounds that activate antioxidant pathways, such as Nrf2 activators, among others. The simultaneous modulation of the circadian cycle and oxidative stress in vascular dementia may offer a more effective therapeutic approach. The goal of these treatments is to improve cognitive function, slow the progression of the disease, and treat any underlying conditions that may contribute to the development of vascular dementia [70].

#### 5.1.1. Melatonin

Melatonin is a hormone synthesized in the body that is responsible for regulating the circadian cycle. It can be administered exogenously and has been shown to have neuroprotective effects on vascular dementia because it increases the concentrations of acetylcholine, norepinephrine, and dopamine in the hippocampus [13]. Melatonin also effectively reduces markers of oxidative stress and increases levels of antioxidant factors, thereby reducing ROS overload and oxidative damage. Additionally, it has anti-inflammatory activity, which limits the production of excessive amounts of ROS [71]. It produces antioxidant and anti-inflammatory effects by modulating several signaling pathways including JNK, NF-kB, HIF-1a, and Nrf2. It also increases the expression of protein 30 (SMP30) and osteopontin (OPN), markers of senescence, which have neuroprotective effects and inhibit apoptosis [63,72]. 

Different studies have shown that melatonin improves cognitive dysfunction, suppresses oxidative stress and neuroinflammation, and restores brain-derived neurotrophic factor (BDNF) levels in venous disease models. Melatonin has antioxidant properties that counteract oxidative stress, mitochondrial damage, and apoptosis, which are factors of aging and neurodegenerative disorders. It can also alleviate synaptic dysfunction and increase the number of dendrites, which are essential for the transmission of neuronal signals and the encoding of information [19,73]. In the study by Wang et al., in an animal model of vascular dementia, it was shown that melatonin, apart from enhancing the efficiency of the circadian rhythm, reduced neuroinflammation and improved cognitive function [74]. Other agents that influence the circadian cycle, such as agomelatine or ramelteon, agonists of the MT1 and MT2 melatonin receptors, have demonstrated positive effects in vascular dementia [75]. These results suggest the modulation of the circadian cycle as a promising therapeutic approach.

#### 5.1.2. Nrf2 Activators

One of the most encouraging molecular targets in the fight against oxidative stress is the transcription factor Nrf2, which plays a central role in regulating the cellular antioxidant response and regulates the expression of genes involved in antioxidant defense and cellular detoxification [76]. Sulforaphane, a compound present in cruciferous vegetables such as broccoli, has demonstrated antioxidant properties by inducing the expression of genes regulated by Nrf2 [77]. Additionally, bardoxolone methyl, an Nrf2 activator, has been investigated in clinical trials to evaluate its efficacy in patients with chronic kidney disease. It has been shown to induce the expression of antioxidant enzymes [78]. Studies have suggested that combining agents that regulate the circadian cycle, such as melatonin, with Nrf2 activators may have a synergistic impact on improving symptoms and slowing disease progression [13]. The application of these agents in vascular dementia is under active investigation.

#### 5.1.3. GLT2 Inhibitors

Sodium-glucose cotransporter 2 (SGLT2) inhibitors or “flozins,” in addition to reducing blood glucose, have a potential effect on atherosclerosis and cognitive impairment. A study in patients with diabetes using empagliflozin, an SGLT2 inhibitor, found a significant decrease in complex intimal media thickness (CiMT), which is a marker of early-stage atherosclerosis [79]. Likewise, atherosclerotic lesions in intracranial and extracranial arteries have been related to cognitive impairment and dementia. In this sense, SGLT2 inhibitors have been demonstrated to reduce vascular inflammation, mitigate oxidative stress, and improve endothelial dysfunction, which contributes to their anti-atherosclerotic effect and, therefore, could have a beneficial effect on vascular health. These effects may contribute to protecting against cognitive decline [33,79,80]. This field of study represents an area of growing interest in the search for therapeutic strategies that address both cardiovascular and cognitive health in patients with chronic medical conditions like diabetes.

#### 5.1.4. Nicotinamide

Nicotinamide is a vitamin that plays a crucial role in metabolic dysfunction and diseases such as diabetes mellitus. It is the amide form of vitamin B3 (niacin) and can be obtained through synthesis in the body or as a source and dietary supplement. Nicotinamide participates in the energy metabolism, synthesis, and repair of DNA through its conversion into NAD. It is also necessary for the synthesis of nicotinamide mononucleotide (NMN), which participates in the production of NAD and NADP through various enzymatic reactions. The role of nicotinamide in the mTOR pathways is important, as it regulates mTOR activity and can inhibit mTORC1 and mTORC2 complexes. The activation of mTOR pathways by nicotinamide may have beneficial effects on insulin secretion, oxidative stress, and glucose homeostasis [19,54,55].

Nicotinamide is closely related to SIRT, a family of NAD+-dependent histone deacetylases. The SIRT1 participates in the regulation of cellular function and survival during metabolic disease; they function through autophagic pathways, which involve the degradation and recycling of cellular components such as mitochondria. The role of nicotinamide in autophagy and SIRT1 may vary depending on certain conditions and may promote cell survival by inducing SIRT1-dependent autophagy and improving cognitive impairment [81,82].

#### 5.1.5. Phenolic Compounds

Phenolic compounds are a group of phytochemicals found in various foods and beverages of plant origin. They are known for their antioxidant properties and researchers have studied their potential health benefits. They work by eliminating free radicals and reducing oxidative stress in the body, which can help protect against various diseases. They have anti-inflammatory effects, improve the lipid profile, reduce blood pressure, and improve endothelial function, which are essential factors related to cardiovascular health [83,84].

Phenolic compounds have been investigated in relation to dementia and have demonstrated their potential to reduce neuroinflammation, oxidative stress, and Aβ accumulation, which are critical factors in the development and progression of dementia. Some phenolic compounds, such as resveratrol, have been found to activate sirtuins involved in cellular processes that may have neuroprotective effects, and modulate central and peripheral biological rhythms, altering the expression of clock genes [82]. The administration time of phenolic compounds can have different effects, such as changes in clock gene expression and metabolite levels. Seasonal rhythms can also affect the bioavailability and effects of phenolic compounds. For example, exposure to different photoperiods significantly affects the bioavailability of red grape polyphenols; the out-of-season consumption of fruits rich in phenolic compounds can alter parameters related to appetite signaling pathways and lipid and glucose homeostasis, as well as modulate the expression of clock genes, such as *SIRT1*, *PER1*, *PER2*, *BMAL1*, and *Rev-Erb* [9,82,84,85].

## 6. Conclusions

Oxidative stress is related to the circadian cycle through various mechanisms; however, the main one is through the generation and elimination of ROS, which exhibit circadian rhythms in different tissues and organs of the body, including the brain. There is a lack of studies that investigate the peripheral and molecular markers of the circadian rhythm in patients with vascular dementia. Nevertheless, patients with cerebrovascular disease exhibit alterations in melatonin, cortisol, temperature, blood pressure, and the expression of the clock genes *BMAL1*, *CLOCK*, *PER*, and *CRY*. The dysregulation of these markers negatively affects cognitive function and may contribute to the development or worsening of dementia symptoms.

In such patients, vascular dementia restricts our understanding. Additionally, markers of oxidative stress, such as lipid oxidation products (like isoprostanes and malondialdehyde), oxidized proteins, DNA damage (assessed by 8-OHdG), and the decreased activity of antioxidant enzymes such as SOD, CAT, and GPX, constitute palpable evidence of the redox imbalance that underlies pathogenesis. This complex network of oxidative stress markers and subsequent circadian dysregulation intertwine integrally in the cognitive progression of vascular dementia. It is essential to conduct more comprehensive research to unravel the peripheral and molecular mysteries of circadian rhythms, thereby enriching our understanding and facilitating the development of more precise therapeutic strategies that simultaneously address the circadian component and oxidative stress. These promising efforts are seen as a path toward new perspectives in the clinical management of this complex neurodegenerative condition.

## Figures and Tables

**Figure 1 ijms-25-04401-f001:**
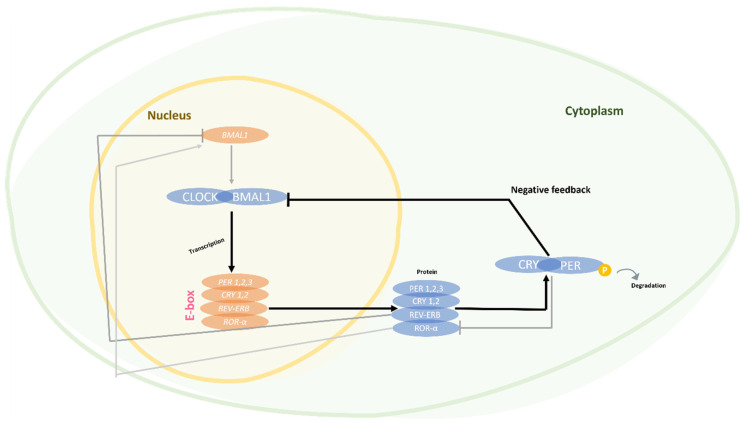
Molecular circadian clock. Black lines: main negative loop; molecularly, we have four main genes that regulate the cycle through negative feedback; the CLOCK and BMAL1 proteins form a heterodimer, acting as a transcriptional activator of *PER* and *CRY*; the PER and CRY proteins are translocated to the cytoplasm and, in turn, they form a heterodimer that negatively inhibits CLOCK and BMAL1. During the day, the PER/CRY heterodimer is elevated, inhibiting CLOCK/BMAL1; at night, it is ubiquitinated, degraded, and allows the cycle to begin. Gray lines: another mechanism that CLOCK/BMAL induces is the transcription of orphan nuclear receptors *Rev-Erb* and *ROR-α*. Rev-Erb, in turn, is an inhibitor of *BMAL1*, and ROR-α is a stimulator; also, the CRY/PER heterodimer inhibits ROR-α.

**Figure 2 ijms-25-04401-f002:**
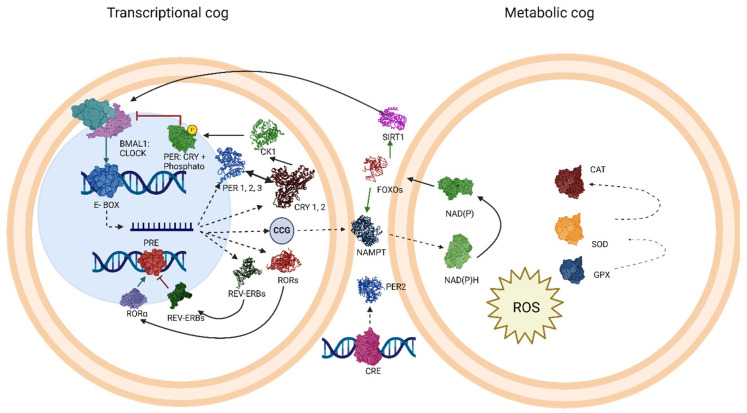
Relationship between oxidative stress and molecular clock. The schematic shows the cycle of the transcriptional cog and metabolic cog. In the TTFL mechanism, the CLOCK/BMAL1 heterodimer binds to promoters (E-BOX), initiating the transcription of *PER* and *CRY* genes. Following transcription, messenger RNA moves to the cytoplasm, facilitating the translation and formation of PER and CRY proteins. These proteins, in a phosphorylated heterodimer form, inhibit their own transcription and that of other CCGs through negative feedback. Over the course of a 24 h day, they degrade, enabling the re-binding of the BMAL1/CLOCK heterodimer. The second loop involves the activation and inhibition of the orphan nuclear receptors related to Rev-Erb and ROR-α, which competitively bind to retinoic acid response elements. Dotted lines represent transcription, while lines with arrows on both sides signify binding and heterodimer formation. Promoters are denoted by green arrows; inhibition is represented by a red line. It is noteworthy that the CRE promoter also drives the transcription of the *PER2* gene. The activation of CRE elements is associated with the binding of proteins such as CREB (cyclic AMP response element-binding protein). FOXOs are involved in the regulation of the gene expression of circadian rhythms. The molecules NADP and NADPH participate in redox processes and energy balance, which can impact the activity of genes and proteins in the TTFL. The NAMPT is involved in the synthesis of NAD+ and may affect the activity of sirtuins, such as SIRT1, which modulate protein acetylation, and may have effects on the activity of components of the TTFL, thereby influencing circadian rhythms. SIRT1 plays a role in circadian regulation and is NAD+-dependent. Changes in NAD+ levels caused by the presence of ROS could affect the activity of sirtuins and, therefore, the regulation of circadian rhythms. SIRT1 plays a role in oxidative stress through the initiation of several downstream effectors, including FOXO transcription factors, as well as the generation of O2, through the donation of electrons from NADPH in a reaction catalyzed by NADPH oxidase. Thus, the initiation of oxidative stress and the general redox state of the cell affects the availability of NAD+ and consequent activity of SIRT1. Finally, both NADH and NADPH have been shown to enhance the DNA binding of the CLOCK/BMAL1 and NPAS2/BMAL1 heterodimers to their transcriptional targets, whereas NAD+ and NADP+ inhibit this activity. Thus, there is a direct link between the redox state of NAD+/NADH and NADP+/NADPH and circadian rhythms’ ROS signaling and it is important for this NAD + NADP+ activation. Abbreviations: COG: clusters of orthologous genes; TTFL: transcription–translation feedback loop; CCG: clock-controlled genes; RORs: retinoic acid-related orphan receptors; https://www.ncbi.nlm.nih.gov/pmc/articles/PMC4750502/ (accessed on 3 March 2024). FOXOs: forkhead box-O; NADP: nicotinamide adenine nucleotide phosphate; NADPH: nicotinamide adenine dinucleotide phosphate; NAMPT: nicotinamide phosphoribosyltransferase; ROS: reactive oxygen species; PRX: peroxiredoxins; SIRT 1: sirtuin 1; CAT: catalase; SOD: superoxide dismutase; GPX: glutathione peroxidase.

**Figure 3 ijms-25-04401-f003:**
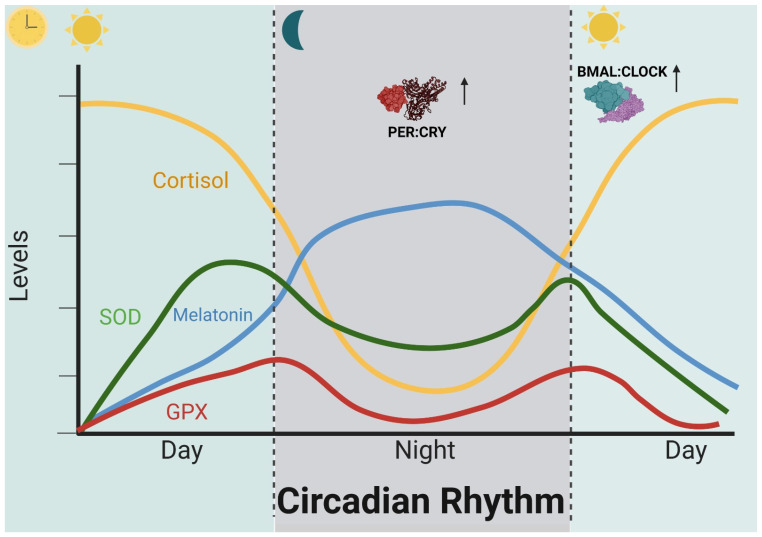
Time-dependent fluctuating levels of some circadian rhythms and oxidative stress markers. In healthy patients, melatonin levels (blue line)begin to rise around 20 h, reach their highest concentration in plasma at 24 h, and then start to decrease; cortisol, on the contrary, begins to rise in the morning and its lowest levels are at night (yellow line), as with the antioxidant enzymes superoxide dismutase (green line) and glutathione peroxidase (red line). Regarding the clock genes, the PER:CRY heterodimer is in higher concentrations during the night and is degraded in the morning, and the CLOCK:BMAL heterodimer is elevated during the morning and is degraded during the night. Abbreviations: SOD: superoxide dismutase; GPX: glutathione peroxidase.

**Table 1 ijms-25-04401-t001:** Changes in molecular and peripheral markers of the circadian cycle in neurodegenerative disease. Overall, the data suggest that patients with neurodegenerative diseases exhibit an alteration of the circadian cycle. Abbreviations: CBT: core body temperature; HD: Huntington’s disease; RBD: rapid eye movement sleep behavior disorder.

	Peripheral Markers	Molecular Markers
Huntington’s disease	Melatonin: reduced levels, delayed phase of secretion, and becomes more delayed with disease progression [17,23,24] Cortisol: some patients have altered levels in the morning. There is a higher amplitude of the rhythm. The greater the progression of the disease, the greater the phase advance, and the higher the plasma levels [17]. CBT: elevated during daytime in presymptomatic HD patients [25].	Disruption of the *mPer2* and *mBmal1* circadian clock genes in the suprachiasmatic nucleus of R6/2 mice [18].
Parkinson’s disease	Melatonin: reduction in amplitude of secretion [17]. Cortisol: higher levels in plasma compared to healthy patients of the same age, and a decreased amplitude of the secretion rhythm [17]. CBT: amplitude of the rhythm was reduced in patients with RBD and dementia with Lewy bodies [26].	Loss of rhythms in *BMAL1* and altered expression of *PER2* and *Rev-Erbα* [19].
Alzheimer’s disease	Melatonin: decreased levels, delayed phase, and reduced amplitude [17]. Cortisol: higher levels in plasma compared to healthy patients of the same age, and phase-advanced rhythm. CBT: delayed phase [27].	Diurnal rhythm pattern was lost in pineal gland. Rhythmic methylation of *BMAL1* was altered in fibroblasts [20].

## Data Availability

The data presented in this study are available on request from the corresponding author due to ethical reasons.

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
