# Peer review of "Modulation of the Circadian Rhythm and Oxidative Stress as Molecular Targets to Improve Vascular Dementia: A Pharmacological Perspective"

_ijms, 2024, doi:10.3390/ijms25084401_

Round 1

Reviewer 1 Report

Comments and Suggestions for Authors

The authors of this article have reviewed the updated information about the important role of the circadian rhythm and oxidative stress in modulating vascular dementia and suggested potential therapeutic approaches. This review article reflects the timely focus on the emerging research of circadian rhythm and oxidative stress in vascular dementia.

However, a few points need to be addressed to improve the current manuscript.

General comments:

1)    In many places, the authors simply listed many references. For instance, the authors wrote that “Dysregulation of these markers such as BMAL1, CLOCK, PER, and CRY, negatively affects cognitive function and may contribute to the development or worsening of dementia symptoms,” (lines 522-523 and many other places) without the detailed underlying mechanisms. Thus, it would be helpful if the authors describe underlying mechanistic explanations in many places.

2)    It would be highly desirable for the authors to include a figure to briefly show the time-dependent fluctuated levels of (A) the critical proteins, such as clock, Bmal, Per, and many antioxidant enzymes, as well as (B) antioxidants, including melatonin, cortisol, and antioxidant vitamins if known, during the 24-h period/day, to help the readers for better understanding of the whole picture. Please indicate the levels of some of these factors in broken lines, if they are known to be decreased in vascular dementia.  

3)    One of the main affected areas of oxidative stress is mitochondria and neuronal cell death. Thus, this reviewer would ask the authors to mention the effects of increased oxidative stress on mitochondrial dysfunction and neuronal cell death in vascular dementia and underlying mechanisms, leading to neuroinflammation and neurodegeneration.  

Minor comments:

1.    Lines 101 and 159: Please describe the definition of Alzheimer’s s-type dementia and general phenotype conditions for the readers.

2.    Lines 439-442: The correct reference number for Wang et al. should have been #58 instead of #55. Thus, this incorrect reference need to be double-checked and corrected.

3.    Lines 455-458: “Bardoxolone” and “Nrf2” words could not be found in the reference #73, which dealt with the chronic kidney disease. Thus, this incorrect reference need to be corrected.

4.    Lines 458-460: “Melatonin” word could not be found in the reference #74. Thus, this incorrect reference need to be double-checked and corrected.

5.    Line 493: The reference should be arranged by the order of their appearance. If this is true, the listed references [77,79] should have been [77,78] while the old reference #78, which comes later (line 501), should be changed to a new reference #79.

Comments on the Quality of English Language

Minor revisions on English expression.

Author Response

Modulation of the Circadian Rhythm and Oxidative Stress as Molecular Targets to Improve Vascular Dementia: A Pharmacological Perspective

Response to Reviewer 1

  1. Summary:

  1. Point-by-point response to Comments and Suggestions

Comments 1: In many places, the authors simply listed many references. For instance, the authors wrote that “Dysregulation of these markers such as BMAL1, CLOCK, PER, and CRY, negatively affects cognitive function and may contribute to the development or worsening of dementia symptoms,” (lines 522-523 and many other places) without the detailed underlying mechanisms. Thus, it would be helpful if the authors describe underlying mechanistic explanations in many places.

Response 1: Thank you for pointing this out. As the aim of the review was to understand how circadian rhythm and oxidative stress relate to vascular dementia and identify markers, only in the conclusion is it mentioned that altered expression of these clock genes, whether upregulated or downregulated, contributes to the development or worsening of dementia symptoms. The specific mechanism of clock genes and dementia progression is discussed from line 182 to 199. It is important to note that there are no studies on peripheral and molecular markers in patients with vascular dementia, so this article paves the way for the start of new research projects.

Comments 2:  It would be highly desirable for the authors to include a figure to briefly show the time-dependent fluctuated levels of (A) the critical proteins, such as clock, Bmal, Per, and many antioxidant enzymes, as well as (B) antioxidants, including melatonin, cortisol, and antioxidant vitamins if known, during the 24-h period/day, to help the readers for better understanding of the whole picture. Please indicate the levels of some of these factors in broken lines, if they are known to be decreased in vascular dementia.  

Response 2: We made a figure with the time-dependent fluctuated levels of melatonin, cortisol, superoxide dismutase (SOD), glutathione peroxidase (GPX), and the principal heterodimers of clock genes in normal subjects. As we mentioned in the previous question, there are no studies on patients with vascular dementia. We added the figure in line 458.

Comments 3: One of the main affected areas of oxidative stress is mitochondria and neuronal cell death. Thus, this reviewer would ask the authors to mention the effects of increased oxidative stress on mitochondrial dysfunction and neuronal cell death in vascular dementia and underlying mechanisms, leading to neuroinflammation and neurodegeneration. 

Response 3: We agree with this comment. Therefore, we incorporated a new subtopic in section 3, in line 310-349.

Comments 4:  Lines 101 and 159: Please describe the definition of Alzheimer’s s-type dementia and general phenotype conditions for the readers.

Response 4: We add the definition of Alzheimer’s type dementia and the difference with vascular dementia in line 168-180.

Comments 5: Lines 439-442: The correct reference number for Wang et al. should have been #58 instead of #55. Thus, this incorrect reference need to be double-checked and corrected.

Response 5: Thank you for the observation, there was a problem with the arrangement of authors and that is why the name of another author appeared at the beginning, but appropriate changes were made to the text in line 548.

Comments 6: Lines 455-458: “Bardoxolone” and “Nrf2” words could not be found in the reference #73, which dealt with the chronic kidney disease. Thus, this incorrect reference need to be corrected.

Response 6: The reference was modified in the text to the correct reference in line 562.

Comments 7:  Lines 458-460: “Melatonin” word could not be found in the reference #74. Thus, this incorrect reference need to be double-checked and corrected.

Response 7: The reference was modified in the text to the correct reference in line 564.

Comments 8: Line 493: The reference should be arranged by the order of their appearance. If this is true, the listed references [77,79] should have been [77,78] while the old reference #78, which comes later (line 501), should be changed to a new reference #79.

Response 8: Thank you for the observation, appropriate changes were made to the text in lines 598 and 606.

  1. Response to Comments on the Quality of English Language
    1. Minor revisions on English expression.
      1. We send the article to correct the English

Reviewer 2 Report

Comments and Suggestions for Authors

The manuscript by Trujillo-Rangel et al. discusses the connection between circadian rhythm and oxidative stress with the development and progression of vascular dementia. The molecular perspective presented is novel and intriguing, indicating regulatory points in this interaction both collectively and individually. However, except for the last section (5. Mechanism of Action), this work is confusing overall, with mixed concepts and numerous repetitions. Additionally, grammar, punctuation, etc., need to be revised. Major points should be addressed:

- Section 2: Circadian Rhythm and Vascular Dementia: the text in this section is confusing and hard to read. There should be a certain order when addressing different points inside this section, such as distinguishing concepts for peripheral markers, circadian clock genes, neurodegenerative disorders, and vascular dementia; currently, these are all mixed together. For example, the paragraph from lines 105 to 131 should come immediately after describing the physiology of circadian rhythms, at line 81. Also, the discussion of patients, scattered across various paragraphs (starting at lines 81, 93, 138), should be unified as there is repetition of concepts and it is very disorganized.

- The paragraph on signal transduction pathways involved in vascular dementia should not be an independent section.

- The title of the table and the figure should not be in upper case.

- The description of the autophagy process is unnecessary for the understanding of the main points in this review.

- The paragraph from line 366 to line 391 should be included before the paragraph at line 322. Ensure it is not repetitive and avoid repeating the name and abbreviation of factors (e.g., as seen with casein kinase 1).

- The paragraph from lines 339-345 is repeated in the paragraph at lines 392.

- The sentences at lines 450 and 451 are nearly identical and should be modified.

Comments on the Quality of English Language

English should be extensively edited and grammar, punctuation, repetitive sentences etc., need to be revised

Author Response

Modulation of the Circadian Rhythm and Oxidative Stress as Molecular Targets to Improve Vascular Dementia: A Pharmacological Perspective

Response to Reviewer 2

  1. Summary:

  1. Point-by-point response to Comments and Suggestions

Comments 1: Section 2: Circadian Rhythm and Vascular Dementia: the text in this section is confusing and hard to read. There should be a certain order when addressing different points inside this section, such as distinguishing concepts for peripheral markers, circadian clock genes, neurodegenerative disorders, and vascular dementia; currently, these are all mixed together. For example, the paragraph from lines 105 to 131 should come immediately after describing the physiology of circadian rhythms, at line 81. Also, the discussion of patients, scattered across various paragraphs (starting at lines 81, 93, 138), should be unified as there is repetition of concepts and it is very disorganized.

Response 1: The paragraphs were arranged. First, it mentions the physiology of the circadian rhythm of peripheral and molecular markers, then it mentions how these markers are in neurodegenerative diseases, then it emphasizes that there are no studies of these markers in patients with vascular dementia, but it discusses how they are found in some diseases related to vascular dementia, such as cerebrovascular disease or in studies in rat models with cardiovascular diseases. We end with the paragraph mentioning how little is known about the molecular and peripheral markers related to the development of dementia progression and the risk factors that predispose to dementia.

Comments 2:  The paragraph on signal transduction pathways involved in vascular dementia should not be an independent section.

Response 2: The corresponding paragraph was removed for clarity.

Comments 3: The title of the table and the figure should not be in upper case.

Response 3: We modified the upper-case title of the figures and table.

Comments 4:  The description of the autophagy process is unnecessary for the understanding of the main points in this review.

Response 4: Thank you for pointing this out, we removed it from the text.

Comments 5: The paragraph from line 366 to line 391 should be included before the paragraph at line 322. Ensure it is not repetitive and avoid repeating the name and abbreviation of factors (e.g., as seen with casein kinase 1)

Response 5: Thank you for the observation, appropriate changes were made to the text in lines 422-450.

Comments 6: The paragraph from lines 339-345 is repeated in the paragraph at lines 392

Response 6: Thank you for the observation, we agree, therefore appropriate changes were made to the text in lines 493-506.

Comments 7:  The sentences at lines 450 and 451 are nearly identical and should be modified.

Response 7: Thank you for the observation, we agree, it was repetitive, therefore we removed it.

  1. Response to Comments on the Quality of English Language
    1. English should be extensively edited and grammar, punctuation, repetitive sentences etc., need to be revised
      1. We send the article to correct the English